# Quantified Vascular Calcification at the Dialysis Access Site: Correlations with the Coronary Artery Calcium Score and Survival Analysis of Access and Cardiovascular Outcomes

**DOI:** 10.3390/jcm9051558

**Published:** 2020-05-21

**Authors:** Hyunsuk Kim, Bom Lee, Gwangho Choi, Ho Yong Jin, Houn Jung, Sunghyun Hwang, Hojung Yoon, Seok hyung Kim, Hoon Suk Park, Jongseok Lee, Jong-Woo Yoon

**Affiliations:** 1Department of Internal Medicine, Chuncheon Sacred Heart Hospital, Chuncheon 24253, Korea; keeee@hanmail.net (H.K.); spring150162@hallym.or.kr (B.L.); sindowhikaru@hallym.or.kr (G.C.); ness9@hallym.or.kr (H.Y.J.); honey1357@hallym.or.kr (H.J.); med2011@hallym.or.kr (S.H.); memory493@hallym.or.kr (H.Y.); thrupy7@hallym.or.kr (S.h.K.); 2Division of Nephrology, Department of Internal Medicine, Eunpyeong St. Mary’s Hospital, School of Medicine, The Catholic University of Korea, Seoul 07345, Korea; cttailor@catholic.ac.kr; 3Department of Business Administration, Hallym University College of Business, Chuncheon 24253, Korea; ljs1844@hallym.ac.kr

**Keywords:** vascular calcification, tomography, X-Ray computed, coronary vessels, renal dialysis

## Abstract

Vascular calcification is a major contributor to mortality in end-stage renal disease (ESRD) patients. In this study, we investigated whether there was a correlation between the coronary artery calcium score (CACS) and the vascular calcification score (VCS), and whether higher VCS increased the incidence of interventions and major adverse cardiac and cerebrovascular events (MACCE). ECG-gated CT, including vascular access and the coronary vessel, was taken. CACS and VCS were calculated by the Agatston method. A comparison of CACS and survival analysis according to VCS groups was performed. Using a cutoff of VCS = 500, 77 patients were divided into two groups. The vintage was significantly older in the higher VCS group. The median CACS was higher in the higher VCS group (21 [0, 171] vs. 552 [93, 2430], *p* < 0.001). The hazard ratio (HR) for interventions and MACCEs in the higher VCS group increased by 3.2 and 2.3, respectively. Additionally, a longer duration of hemodialysis and higher magnesium levels (>2.5 mg/dL) showed lower HRs for interventions (<1). We quantified VCS and found that it was associated with the CACS. Additionally, higher VCS increased the risk of access interventions and MACCE. VCS of the access site may be suggested as a biomarker to predict ESRD patients.

## 1. Introduction

Cardiovascular events are known to be a major cause of mortality in patients with chronic kidney disease [1]. Therefore, it is important for clinicians to assess risk factors for cardiovascular mortality in these patients. In particular, vascular calcification (VC) is known to be a major contributor to mortality and morbidity in end-stage renal disease (ESRD) patients [2,3].

Additionally, the coronary artery calcification score (CACS) is recognized as a strong predictor of cardiovascular mortality in ESRD patients, as well as in the general population [4,5]. In the Chronic Renal Insufficiency Cohort study, when CACS was measured on computed tomography (CT) scans, an increase in the logarithm of CACS by one standard deviation was significantly associated with a 40% higher risk of cardiovascular disease. Additionally, cardiovascular mortality was successfully predicted with CACS measured on an electron-beam CT in ESRD patients [6]. Therefore, coronary artery CT has been proven to be a useful tool for measuring VC.

The mechanism of VC in CKD patients is distinct from that of VC in patients with other diseases. In patients with CKD, various risk factors contribute to the transformation of vascular smooth muscle cells into chondrocytes or osteoblast-like cells [7]. Furthermore, high calcium and phosphorus burdens are present in patients with CKD–mineral and bone disorder (MBD) [8]. Moreover, expression of substances that can inhibit VC are thought to be decreased in CKD patients [9]. Therefore, the risk of VC is higher in CKD patients than in the general population. In addition, the prevalence of VC is two to five times higher in ESRD patients than in control patients of the same age [10].

Generally, calcification occurs less frequently in vessels in the arm than in vessels in other areas (e.g., vessels in the leg or the coronary arteries) [11]. Normally, VC is rare in the veins, but it does occur at vascular access sites in ESRD patients. This phenomenon can be explained by arterialization of the remodeled vein of the vascular access due to the high levels of shear stress from strong blood flow [12]. VC of the vascular access sites can be a marker of systemic VC, and it is related to insufficient dialysis and mortality [2]; furthermore, it may also be associated with vascular access failure [13].

A previous study found a difference in patients’ survival rates according to the presence of calcification of the vascular access using plain X-ray examinations. In that study, 23% of patients presented with VC, and there was a significant difference in mortality between patients with and without calcification, a finding that demonstrated the clinical significance of vascular access calcification in ESRD patients [2]. However, a limitation of that study is that it did not quantify VC. Additionally, another study compared VC observed using contrast CT and clinical data. In that study, the contrast-enhanced vessels were divided into the aorta, subclavian artery, carotid artery, and the arterial or venous part of the vascular access, but clinical correlations were not reported [14]. 

However, little research has investigated the correlation of VC at the vascular access site with the CACS, which is known to be a strong predictor of CV risk. Similarly, few studies have focused on the clinical significance of VC at the vascular access in ESRD patients.

Therefore, this study quantified VC in ESRD patients using the Agatston method to investigate its correlation with the CACS. The relationships of VC with vascular access survival, the hospitalization rate, and major adverse cardiac and cerebrovascular events (MACCE) were also investigated. Furthermore, we sought to analyze other risk factors for vascular access failure and MACCE within the framework of VC at the vascular access site. 

## 2. Methods

### 2.1. Patients

Seventy-eight patients who underwent outpatient hemodialysis for at least 3 months were included in the study. CT scans were used to determine the CACS, and laboratory parameters, including hemoglobin, cholesterol, albumin, ferritin, CKD-MBD parameters, and single pool (sp)Kt/V were measured. The medical records of the patients from January 1985 to May 2019 were retrospectively reviewed. Information was collected on patients’ medical history, including ESRD etiology, comorbidities, blood pressure, details of vascular access, hemodialysis vintage, ultrafiltration volume, height, weight, and dry weight. Clinical outcomes, including hospitalization, MACCE, and mortality were reviewed.

MACCE was defined as nonfatal cardiac arrest, acute myocardial infarction, heart failure, new onset arrhythmia (atrial flutter, atrial fibrillation, second- or third-degree atrioventricular block), angina, stroke (embolic, thrombotic, or hemorrhagic), cardiovascular death, or cerebrovascular death.

The male-to-female ratio was 44:34, and after the exclusion of one patient with a femoral arteriovenous graft, 77 patients were ultimately analyzed (Appendix A). 

### 2.2. Protocols of CT and Calculation of CACS and VCS

CT scans were taken from the chest to the pelvis with a slice thickness of 3 mm. Electrocardiographically gated non-contrast CT was performed to obtain measurements suitable for calculating the CACS. The scan parameters were 120 kVp and 135 mAs. The radiation doses were quantified by the median direct dose profile integral (DPI), which had a value of roughly 500 DPI, which is a similar or lower amount compared to the radiation from a non-contrast CT scan from the chest to the pelvis.

We used the Agatston method to calculate the CACS and VCS by multiplying the area of each calcified lesion by a weighting factor corresponding to Hounsfield units. Figure 1 presents a case in which the CACS and VCS were 558 and 4838, respectively, corresponding to a fairly high level of VC. 

The 77 patients were divided into two groups according to their VCS score (≤500 or >500). A cut-off of 500 was chosen to include the top 40% of subjects in this study, because a CACS over 400 is indicative of high risk for major cardiovascular events, and approximately 40% of the subjects in this study had a CACS over 400 [15]. The Agatston method was implemented using a semi-automatic program. First, we designated a calcified lesion, as the program is capable of annotating a continuous lesion, but calibration is necessary to determine whether it properly identifies lesions. The VCS was calculated by summing the calcification score calculated on each slice (Figure 1). Although this technique is somewhat cumbersome, it was possible to measure the VCS of each patient in 10–30 min.

An electrocardiographically gated non-contrast, chest-abdomen-pelvis computed tomography was taken. The Agatston method was implemented using a semi-automatic program. We designated a calcified lesion, as this program is capable of annotating a continuous lesion, but it must be calibrated to determine whether lesions are properly identified. The VCS was calculated by summing the calcification score calculated on each slice. It took 10 to 30 min to measure the VCS in each patient. Abbreviations are as follows: CT, computed tomography; DPI, direct dose profile integral; A-P, abdominal-pelvis; CACS, coronary artery calcium score; and VCS, vascular calcification score.

### 2.3. Statistical Analysis

The data are expressed as median (interquartile range). Variables were compared using the Mann–Whitney U test. Since interventions, hospitalizations, and MAACE can occur multiple times in a single patient and each event is not independent, the Prentice, Williams, and Peterson total time (PWP-TT) survival analysis model was used. Possible confounders and clinically important factors were included in the univariate analysis, and all variables with P-values less than 0.05 in the univariate analysis were included in the multivariate models. All analyses were performed using SPSS 23.0 (IBM Corp., Armonk, NY, USA) and Stata version 15 (StataCorp LLC, College Station, TX, USA). 

## 3. Results

### 3.1. Baseline Characteristics of the Subjects

The flow chart of the subjects is presented in Appendix A. Table 1 shows the subjects’ baseline demographic and clinical characteristics. There were 46 (59.7%) patients with a VCS ≤ 500 and 31 (40.3%) with a VCS > 500. Sex, age, and the etiology of ESRD between the two groups did not show significant differences. Similarly, no statistically significant difference in blood pressure was observed between the two groups.

Turning to hemodialysis-related parameters, the higher VCS group showed a lower prevalence of native arteriovenous fistulas (VCS ≤ 500 vs. VCS > 500: *n* = 43 [93.5%] vs. *n*=23 [74.2%]; *p* = 0.018) and a longer hemodialysis vintage (median [IQR]: 36.6 [20.9, 60.2] months vs. 89.1 [46.8, 137.6] months; *p* < 0.001). However, no significant differences were found in ultrafiltration volume, height, dry weight, spKt/V, or the urea reduction ratio.

The higher VCS group had significantly lower albumin levels (median [IQR]: 3.8 [3.6, 3.9] g/dL vs. 3.6 [3.3, 4.0] g/dL; *p* = 0.043). Levels of other nutritional markers, such as total cholesterol, triglyceride, and ferritin levels also tended to be lower in the higher VCS group. No significant between-group differences were found for CKD-MBD parameters, such as calcium, phosphorus, or intact parathyroid hormones. Magnesium levels (reference, 1.5–2.5 mg/dL) were also not significantly different between the two groups (Table 1). 

### 3.2. CACS in the Two Groups Defined by VCS

The median VCS of all subjects was 307, and the median CACS was 101. The subjects were divided into two groups according to a cut-off value of VCS of 500 (≤500 or >500, respectively). In the lower VCS group, the median VCS was 144, whereas it was 1058 in the higher VCS group. Moreover, there were 17 (37.0%) patients with a CACS of 0 in the lower VCS group, while only two such patients (6.5%) were found in the higher VCS group. In the lower VCS group, the median CACS was 21, but in the higher VCS group, it was 552, constituting a significant difference (Table 2, Figure 2A). Therefore, it was clear that the higher VCS group had a higher CACS observed on CT. Furthermore, 17% of the subjects in the lower VCS group had a CACS over 400, compared to 61% in the higher VCS group (Figure 2B).

Plain arm X-rays were taken to ensure the sensitivity of VC. Using CT, VC was visible in all 77 patients, but VC was detected in 69 (89.6%) patients on an arm X-ray. Therefore, it was thought that a considerable amount of VC seen on CT can be observed even on an X-ray examination, but X-ray examinations could not detect all VC. In the lower VCS group, VC was detected in 38 patients (82.6%), while VC was observed in all patients in the higher VCS group on plain X-rays. The differences between the two groups were notable (Table 2). 

### 3.3. Prevalence of Events in the Two Groups Defined by VCS

Table 3 presents the prevalence of events. In the higher VCS group, 23 (74.2%) subjects underwent interventions such as percutaneous transluminal angioplasty or surgical thrombectomy or revision. In contrast, only nine (19.6%) of the lower VCS group experienced interventions. The number of annual interventions was also higher in the higher VCS group (lower: 0 [0.0] vs. higher: 0.46 [0, 1.25]; *p* < 0.001). No significant differences between the two groups were found in terms of the number and duration of admissions or the occurrence or number of MACCE.

In the PWP-TT survival analysis mode, the hazard ratio [HR] for interventions in the higher VCS group indicated a 3-fold higher risk (higher VCS group: 3.232 [1.300, 8.034]). A longer duration of hemodialysis was associated with a lower HR for interventions (0.993 [0.989, 0.998]), which implies that more interventions might have been performed in the early phase after starting access. Furthermore, patients with an arteriovenous graft had a two-fold higher likelihood of receiving an intervention (2.309 [1.158, 4.605]). Interestingly, a significantly reduced HR for interventions was found in patients with higher magnesium levels (0.488 [0.281, 0.849]) than the normal cut-off of 2.5 mg/L (Figure 3). The distribution of magnesium levels in the subjects is presented in Appendix A). 

In the higher VCS group, the cumulative incidence of interventions was higher, with a hazard ratio (HR) of 3.232. Longer HD duration decreased the HR, and AVG increased the HR for an intervention. Interestingly, higher magnesium levels decreased the HR for an intervention. Abbreviations are: VCS, vascular calcification score; VC, vascular calcification; HR, hazard ratio; HD, hemodialysis; AVG, arteriovenous graft; AVF, arteriovenous fistula; and Mg, magnesium.

In Figure 4, we plotted the incidence of MACCE in both groups using Cox regression, and found no significant between-group difference. However, the PWP-TT model yielded an HR of MACCE of 2.3 in the group with a VCS > 500. Additionally, survival analysis did not show a significant between-group difference in hospitalization or mortality. 

A graph of MACCE generated by Cox regression did not show a significant difference between the two groups. However, in the PWP-TT model, the hazard ratio (HR) was higher (2.309) in the higher VCS group. Abbreviations are: VCS, vascular calcification score; VC, vascular calcification; HR, hazard ratio.

To summarize our results, it was confirmed that VC can be quantified using the Agatston method, and that the VC calculated using CT was closely correlated with the CACS. In addition, higher VC increased the risk of an access intervention, and the risk of an intervention was higher in cases with an arteriovenous graft or low magnesium levels. Furthermore, it was shown that higher VC was associated with an increased risk of MACCE. 

## 4. Discussion

The motivation for this study was derived from a question regarding the clinical significance of VC at the vascular access site in ESRD patients. The hypothesis of this study was that, because the VC of the arm could function as an indicator similar to the CACS, a higher VCS would predict a higher CACS. Moreover, we speculated that cardiovascular mortality would be elevated in the higher VCS group, as the CACS is linked to cardiovascular mortality. As expected, a positive correlation was found between the CACS and VCS, and the higher VCS group had a higher frequency of interventions. Although no significant difference was found in the frequency of hospitalization or mortality according to the VCS, the higher VCS group showed a significant difference in MACCE.

Since the blood vessels are systemically connected, it is an interesting question whether local VC is an indicator of systemic VC. In fact, it is difficult to examine VC throughout the entire body and to determine relationships between events of interest and VC. However, both in previous research [16] and in this study, it was observed that VC in the arm was associated with hard outcomes, such as MACCE, similarly to what has been found for the CACS.

The importance of this study is that we were able to visualize and quantify VC, and the quantified VC of vascular access showed correlations with outcomes such as vascular interventions or MACCE. We all know that VC is deleterious, but few studies have sought to visualize it. Therefore, an advantage of our study is that we enabled a more precise analysis by quantifying VC. Without considering our VC quantification as a future possibility of diagnostic testing, it is interesting to note that this study validated the importance of VC of the vascular access. Additionally, VC that was undetectable on plain X-rays was observed on CT. The VCS calculated based on CT was correlated with the CACS. Therefore, CT could provide more accurate and quantified data than was possible with plain X-rays.

It is very interesting that the VC of the arm influenced the intervention rate at the vascular access. VC is not well known to reduce access survival in HD patients. Lyu et al. reported that lower fetuin-A levels and higher levels of osteopontin and bone morphogenetic protein-7, as markers of VC, were associated with vascular failure [17]. The calcification score obtained using the Doppler ultrasound can predict lower blood access flow, which is a predictive sign of access failure [18], and Baktiroglu et al. stated that the poor outcomes of vascular access in diabetes are due to arterial disease [19]. To sum up, these previous studies indirectly suggested a relationship between VC and access survival. We added more direct evidence for this possibility by showing that the higher VC group had a higher HR for interventions. This report will be helpful for interventional nephrologists.

However, the limitations of this study include uncertain causality due to the retrospective analysis of data, as well as the small sample size, which reflects the fact that patients from a single hospital were analyzed. Furthermore, the outcomes may not be more broadly generalizable to CKD patients than ESRD patients. The results presented herein may be specific to ESRD patients because the blood flow in ESRD patients is abnormally high at the vascular access, which involves different mechanisms than those in other physiological vessels. Additionally, although the mechanisms and clinical implications may differ between arteries and veins in ESRD patients, it was not possible to distinguish between arteries and veins in this study due to the use of non-contrast CT. Furthermore, this study did not consider relationships with conventional risk factors for cardiovascular mortality, such as diabetes, smoking, and dyslipidemia. This was because of the small sample size and the fact that dyslipidemia was relatively well-controlled with medications in most subjects. In addition, CKD-MBD parameters were not different between the VCS groups, which may have been due to well-controlled management using medications. Lastly, it was not possible to distinguish between intimal and medical calcification in ESRD patients in this study, which would require the use of ultrasonography or vascular biopsy in further research. 

VCS calculation may be somewhat cumbersome, and the usability of this score may therefore be limited. However, if the process of VCS measurement through CT can be simplified with artificial intelligence or other advances, it can provide important information about the patient's vascular survival and outcome. This point is considered to be valuable for future research, as it implies that basic guidelines for VCS detection can help clinicians and patients. Furthermore, the finding of the present study that the evaluation using plain X-rays showed evident results, which were also correlated with the VCS, can be translatable to clinical use more readily. However, the use of plain X-rays for this purpose also requires further testing with large cases.

Furthermore, additional research should investigate the role of magnesium, as this study showed a lower prevalence of interventions in patients with higher magnesium levels. Magnesium is known to bind phosphate and to delay calcium phosphate crystal growth, and it may regulate vascular smooth muscle cell transdifferentiation toward an osteogenic phenotype [20]. It is a novel finding, with interesting implications, that higher magnesium levels decreased the HR for interventions. Some studies have reported that higher magnesium levels were associated with better hard outcomes [21,22], making it necessary to reconsider whether the normal range of magnesium should be applied in ESRD patients in order to enhance their health outcomes and to prevent access interventions. 

In conclusion, although this study had a small number of subjects, it demonstrated a correlation between VC of the access arm and the CACS. Moreover, VC increased the incidence of access interventions and MACCE. 

## Figures and Tables

**Figure 1 jcm-09-01558-f001:**
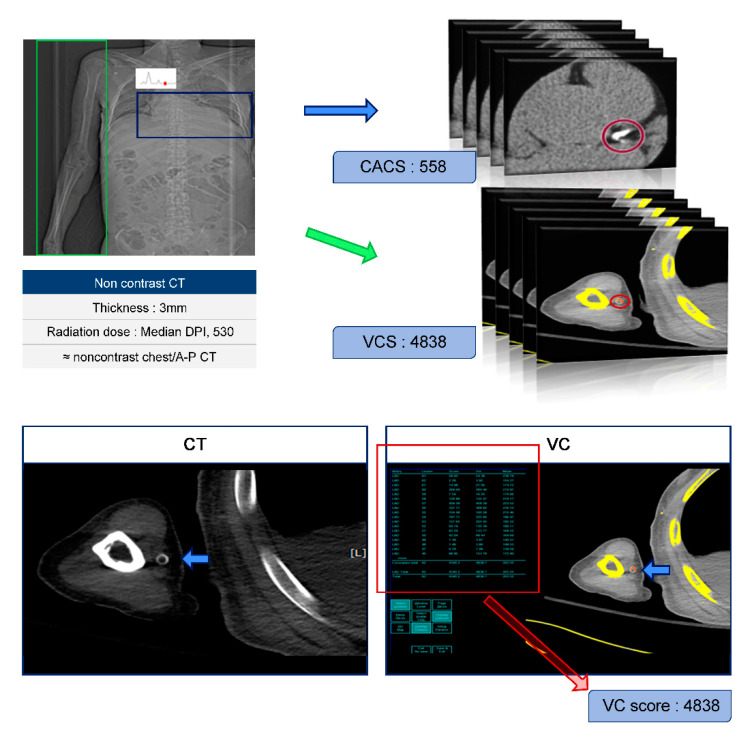
How to calculate the vascular calcification score (VCS).

**Figure 2 jcm-09-01558-f002:**
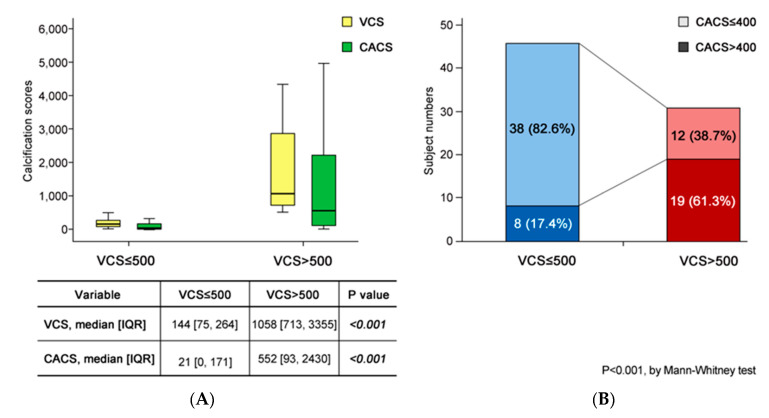
VCS and CACS in the two VCS groups A. Median VCS and CACS. B. Distribution of CACS > 400. (**A**) The median CACS of the lower (≤500) VCS group was 144, and that of the higher (>500) group was 1058. The CACS was different between the two groups (21 vs. 552). (**B**) There were eight (17%) patients in the lower VCS group, and 10 (61%) in the higher VCS group. Abbreviations: VCS, vascular calcification score; CACS, coronary artery calcium score.

**Figure 3 jcm-09-01558-f003:**
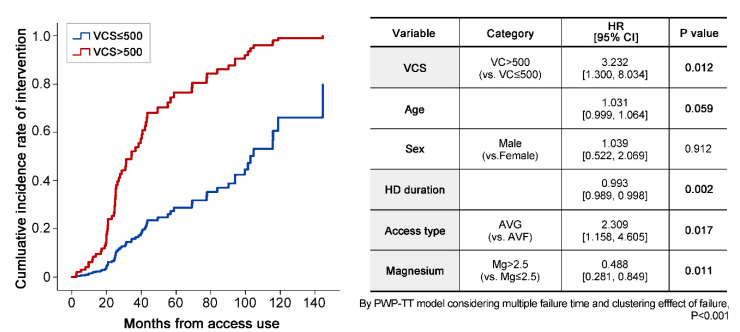
Incidence of interventions in the two VCS groups.

**Figure 4 jcm-09-01558-f004:**
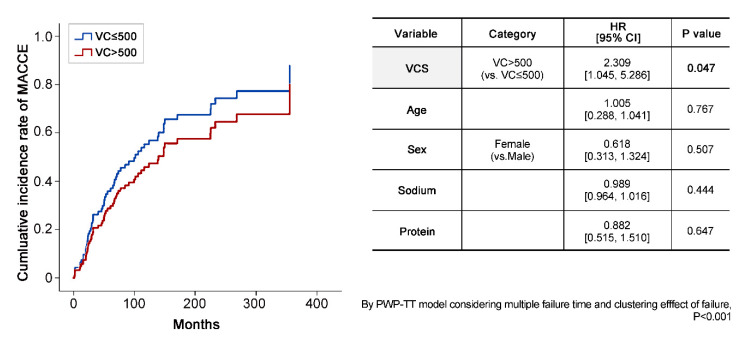
The incidence of MACCE in the two VCS groups.

**Table 1 jcm-09-01558-t001:** Baseline characteristics.

Variable	VCS ≤ 500	VCS > 500	*p*-Value
N (%)	46 (59.7)	31 (40.3)	
Male sex, *n* (%)	23 (50.0)	21 (67.7)	0.564
Age, years, median [IQR]	64.3 [57.0, 74.0]	63.0 [55.0, 67.0]	0.125
Etiology, *n* (%)			0.836
Diabetes	23 (50)	17 (54.8)	
HTN	17 (37)	9 (29.0)	
CGN	0 (0)	1 (3.2)	
PKD	2 (4.3)	0 (0)	
Others	4 (8.7)	4 (12.9)	
Baseline comorbidity			
HTN	35 (76.1)	25 (80.6)	0.638
Stroke	2 (4.3)	0 (0)	0.243
Myocardial infarction or PAD	0 (0)	0 (0)	
SBP, mmHg, median [IQR]	150 [130, 170]	145 [120, 170]	0.457
DBP, mmHg, median [IQR]	70 [70, 80]	70 [60, 80]	0.308
AVF (vs. AVG), *n* (%)	43 (93.5)	23 (74.2)	0.018
HD vintage, months, median [IQR]	36.6 [20.9, 60.2]	89.1 [46.8, 137.6]	<0.001
UF volume, kg, median [IQR]	2.1 [1.5, 3.0]	2.2 [1.5, 3.0]	0.569
Height, cm, median [IQR]	161.7 [154.9, 169.1]	164.0 [154.0, 170.5]	0.884
Dry weight, kg, median [IQR]	60.5 [53.8, 70.1]	57.5 [48.0, 61.5]	0.149
spKt/V, median [IQR]	1.74 [1.49, 2.09]	1.81 [1.63, 2.10]	0.399
URR, %, median [IQR]	77.0 [71.8, 81.3]	77.7 [74.3, 82.3]	0.443
Hb, g/dL, median [IQR]	10.4 [9.8, 11.1]	10.2 [9.6, 10.9]	0.352
Uric acid, mg/dL, median [IQR]	7.2 [6.3, 8.6]	6.9 [5.9, 7.8]	0.253
Total cholesterol, mg/dL, median [IQR]	131.0 [108.5, 147.5]	114.5 [105.3, 139.0]	0.071
Albumin, g/dL, median [IQR]	3.8 [3.6, 3.9]	3.6 [3.3, 4.0]	0.043
Triglycerides, mg/dL, median [IQR]	115.5 [80.0, 182.3]	102.0 [72.0, 116.0]	0.086
Ferritin, ng/mL, median [IQR]	204.1 [115.7, 292.0]	153.5 [67.7, 251.7]	0.072
B2 microglobulin, mg/L, median [IQR]	2.6 [2.4, 2.8]	2.7 [2.4, 2.9]	0.602
25(OH) vitamin D, ng/mL, median [IQR]	21.2 [14.7, 25.7]	19.0 [12.0, 24.4]	0.5
Calcium, mg/dL, median [IQR]	8.1 [7.6, 8.5]	8.3 [7.8, 8.6]	0.255
Phosphorus, mg/dL, median [IQR]	4.8 [3.9, 5.4]	4.7 [3.9, 5.8]	0.762
iPTH, pg/mL, median [IQR]	277.5 [142.3, 423.0]	307.0 [70.2, 596.0]	0.561
Magnesium, mg/dL, median [IQR]	2.6 [2.4, 2.8]	2.7 [2.4, 2.9]	0.602

By the Mann–Whitney test. Abbreviations: VCS, vascular calcification score; IQR, interquartile range; HTN, hypertension; CGN, chronic glomerulonephritis; PKD, polycystic kidney disease; PAD, pheripheral artery disease; SBP, systolic blood pressure; DBP, diastolic blood pressure; AVF, arteriovenous fistula; AVG, arteriovenous graft; HD, hemodialysis; UF, ultrafiltration; sp, single pool; URR, urea reduction ratio; Hb, hemoglobin; iPTH, intact parathyroid hormone.

**Table 2 jcm-09-01558-t002:** CACS and the presence of VC on X-rays in the two VCS groups.

Variable	Total	VCS ≤ 500	VCS > 500	*p*-Value
N (%)	77	46 (59.7)	31 (40.3)	
VCS, median [IQR]	307 [114, 874]	144 [75, 264]	1058 [713, 3355]	<0.001
CACS = 0, *n* (%)	19 (24.7)	17 (37.0)	2 (6.5)	0.002
CACS, median [IQR]	101 [1, 673]	21 [0, 171]	552 [93, 2430]	<0.001
VC on X-rays, n (%)	77	38 (82.6)	31 (100)	0.015

By the Mann-Whitney test. VCS, vascular calcification score; VC, vascular calcification; IQR, interquartile range; CACS, coronary artery calcification score.

**Table 3 jcm-09-01558-t003:** Incidence of events.

Variable	VCS ≤ 500	VCS > 500	*p*-Value
Intervention (PTA or surgery), *n* (%)	9 (19.6)	23 (74.2)	<0.001
Number of interventions/yr, median [IQR]	0 [0, 0]	0.46 [0, 1.25]	<0.001
Number of admissions/yr, median [IQR]	2.9 [1.7, 5.1]	2.9 [2.0, 6.0]	0.553
Duration of admission/yr, median [IQR]	15.3 [5.6, 26.7]	18.4 [6.7, 54.3]	0.114
MACCE, *n* (%)	19 (41.3)	18 (58.1)	0.149
Number of MACCE/yr, median [IQR]	0 [0, 0.1]	0 [0, 0.1]	0.968

By the Mann-Whitney test. PTA, percutaneous transluminal angioplasty; IQR, interquartile range; MACCE, major adverse cardiac and cerebrovascular events.

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
