# Peer review of "Quantified Vascular Calcification at the Dialysis Access Site: Correlations with the Coronary Artery Calcium Score and Survival Analysis of Access and Cardiovascular Outcomes"

_jcm, 2020, doi:10.3390/jcm9051558_

Round 1

Reviewer 1 Report

Kim et al. investigated the association between quantified vascular calcification at the dialysis access and major adverse cardiac and cerebrovascular events. However, I didn’t found any figures or tables embed in this manuscript. Please resubmit the manuscript again.

Author Response

To reveiwer 1.

I am very grateful for your reviewing our manscript.

I uploaded the figures, tables and the supplementary data, maybe you only checked the word file and did not find a separate PDF file.
So I send you a word file that contains figures and tables, provided by the editor to us.

Then kindly ask for a positive review. Thank you again in my heart.

Best regards

Hyunsuk Kim

Reviewer 2 Report

Interesting paper and evaluation of vascular calcification and CAC- suggesting link with high VCS associated with higher CAC scores. In ESRD patients, presence can tentatively relate to high CAC and subsequent events. VCS calculation may be somewhat cumbersome as noted in the paper and therefore usability of this score may be limited. However if basic guidelines for detection of VCS can be related or quantified since the evaluation per plain xray was evident and also correlated with VCS-- then that can be translatable to clinical use easier. However that would have to be tested also. Would be good to have units or scale defined on the graphs presented on graphs/figure on page 6.

Author Response

Dear Reviewer 2.

I am very grateful for your reviewing our manscript.

1. VCS calculation may be somewhat cumbersome as noted in the paper and therefore usability of this score may be limited. However if basic guidelines for detection of VCS can be related or quantified since the evaluation per plain xray was evident and also correlated with VCS-- then that can be translatable to clinical use easier. However that would have to be tested also.
-> I am very sympathetic to your comments. As you said, checking VCS for each patient can require considerable labor depending on VCS severity. Therefore, the part you commented on was revised as follows and included in the discussion.

VCS calculation may be somewhat cumbersome and therefore usability of this score may be limited. However, if the VCS measurement process through CT can be simplified with artificial intelligence, etc., it can be important information about the patient's vascular survival and outcome. This part is considered to be valuable for future research. Through this, basic guidelines for detection of VCS can help clinicians and patients. Furthermore, since the evaluation by plain X-ray was evident and also correlated with VCS, then that can be translatable to clinical use easier. However, that would have to be tested also.

2. Would be good to have units or scale defined on the graphs presented on graphs/figure on page 6.

-> Thank you for kindly pointing out our mistakes. Calcification scores and Subject numbers were inserted on the Y-axis, and these two scores were not included because there were no units.

Best regards

Hyunsuk Kim

Round 2

Reviewer 1 Report

Kim et al. investigated the association between quantified vascular calcification at the dialysis access and major adverse cardiac and cerebrovascular events. The research is interesting, but some study concerns should be mentioned.

1. In Table 1, some important baseline characteristics were not included in this study, such as comorbidities of diabetes, hypertension, old myocardial infarction, old cerebrovascular disease, and peripheral artery disease. Since the outcome of this study is major adverse cardiac and cerebrovascular events (MACCE), the baseline level of cardiovascular risk should be evaluated at baseline and considered as confounding factors.

2. The essential factors associated with MAACE was analyzed using univariate Cox regression model. These factors included VCS, Age, Sex, HD duration, Access type, and magnesium. However, several important factors were not included or adjusted, such as diabetes mellitus, hypertension, old myocardial infarction, old cerebrovascular disease, and peripheral artery disease. The association between VCS and incidence of interventions or MACCE could be bias by the baseline cardiovascular risk factors. Therefore, we can interpret VCS could be related to events and consider as markers. However, it’s not independently associated with the outcomes.

3. How to evaluate the cut-off value of magnesium at 2.5 mg/dL? Is there any reference citation?

4. The definition of MACCE or intervention should be clarified in the methods.

5. The author could provide a clinical application for a physician to evaluate the vascular calcification of vascular access. However, CT is more expensive examination but better image resolution than X-ray. So, how to choose, or what is the suggestion based on this study?

6. The evaluation of vascular calcification of vascular access. Would arteriovenous fistula and arteriovenous graft present different image results or clinical applications?
